# Clinical Stakes of Sexual Abuse in Adolescent Psychiatry

**DOI:** 10.3390/ijerph20021071

**Published:** 2023-01-07

**Authors:** Marion Robin, Thomas Schupak, Lucile Bonnardel, Corinne Polge, Marie-Bernard Couture, Laura Bellone, Gérard Shadili, Aziz Essadek, Maurice Corcos

**Affiliations:** 1Département de Psychiatrie de L’adolescent et du Jeune Adulte, Institut Mutualiste Montsouris, 75014 Paris, France; 2Centre de Recherche en Epidémiologie et Santé des Populations, INSERM U1178, Team PsyDev, 94807 Villejuif, France; 3Centre de Soins, d’Accompagnement et de Prévention en Addictologie, Émergence Espace Tolbiac, 75013 Paris, France; 4Laboratoire Interpsy, Université de Lorraine, 54015 Nancy, France

**Keywords:** sexual abuse, adolescence, mental disorders, suicidal attempts, hospitalization

## Abstract

Background: The extent and nature of sexual abuse (SA) and its consequences in psychiatry are still poorly described in adolescence. Objective: This article describes the frequency of SA reported in an adolescent population hospitalized in psychiatry, and assesses its links with the severity of mental disorders and the medical issues of these adolescents. Methods: The study includes 100 patients for whom SA has been mentioned, among all patients aged 13 to 17 years old hospitalized for about 4 years. The characteristics of sexual abuse were correlated with the medical severity of the patients, as well as the number, the duration of their hospitalization(s), and the time until disclosure. Results: The results show the central place of SA in adolescent psychiatry, with a prevalence of 28.5% and a cumulative hospital stay which is five times longer than average. Correlations have been observed between the number of suicide attempts and the number of abuses reported. The medical severity of patients is significantly increased when the named aggressor is an adult. The number of hospitalizations is positively correlated with the number of reported abuses, as well as with the intrafamilial and adult status of the perpetrator. Finally, an early age of onset, repeated abuse, and the intrafamilial nature of the abuse are associated with a longer time to disclosure. Conclusions: The severity of adolescent psychiatric situation is statistically in favor of a history of SA, which should therefore be actively explored during care.

## 1. Introduction

Sexual abuse of under-18s is a major public health problem, with an estimated prevalence of 15–20% for women and 8–10% for men worldwide [1,2,3]. Adolescence is considered to be the period of life most at risk of sexual assault [4,5], and it is also often at this age that the consequences of sexual abuse manifest themselves, through psychic decompensations. Daily practice in adolescent psychiatry regularly includes the management of clinical situations in which sexual abuse is reported, but the specificity of these situations and the resulting care is not described. Child sexual abuse (CSA) is thought to be associated with 47% of mental disorders appearing in childhood and about 30% of disorders appearing in adulthood [6,7,8,9]. However, there is a great lack of descriptive data concerning adolescents hospitalized in psychiatry, who constitute a high-risk population [10].

Moreover, the understanding of the links between sexual abuse and mental disorders in adolescence is very partial: How does sexual abuse relate to different psychiatric symptoms (traumatic symptoms or others)? What are the links between the characteristics of abuse and medical characteristics: Medical severity, number of hospitalizations, number of suicide attempts? CSA is recognized as a risk factor for a variety of mental disorders [11], but the understanding of the mechanisms by which abuse, in its diversity, exerts its effects is still limited [12]. It seems that the nature of sexual abuse influences the severity of psychopathology in adulthood [13]: Several studies have observed that children who have been assaulted more severely [14,15], more frequently [16], for whom the aggression has lasted a long time [17,18], and by an aggressor with a close affective bond [14,19] developed more serious mental disorders. Beyond the characteristics specific to the abuse, other variables, such as the support offered to the young person [20], appear to better predict longitudinal outcomes [21].

This study explores the extent and nature of SA phenomena reported in a population of adolescents hospitalized in psychiatry, whereas studies to date have mainly focused on adult populations [5,22]. The secondary objective is to assess the links between the reported abuse and the characteristics related to the mental health and the care of these adolescents.

## 2. Materials and Methods

The study undertaken is a descriptive study and includes 100 patients hospitalized in a psychiatric crisis unit (average hospital stay of 3 weeks), for whom SA has been evoked. The unit welcomes adolescents with all types of mental disorders. The study was conducted over 4 years (2017–2021). The clinicians in charge of the patients were asked to mention the situations in which sexual abuse had been evoked during the usual hospital care for each hospitalized patient. For the patients involved, data related to sexual abuse, its disclosure, and clinical elements were collected in the medical record after discharge. The study was voluntarily stopped at the 100th record (a sufficiently large sample size for hypotheses and statistical methods and to facilitate representations), which corresponded to 46 months of care and during the hospitalization of a total of 473 adolescents, resulting in an incidence of 21.1%.

Descriptions of interactions of a sexual nature involving contact with the patient’s body (sexual touching, rape) or attacks on the sensory sphere, such as obscene telephone conversations, exposure to pornography, cyber-harassment or exhibitionism are considered as SA.

Data collected for the study are divided into five types of information: -Socio-demographic data: Sex, age, socio-professional category, divorce.-Data related to SA: Nature of SA (without penetration; with penetration), number of SA (single; repeated), age of onset (<13 years; ≥13 years), relationship to the perpetrator (intrafamilial; extrafamilial), age of named perpetrator (<18 years; ≥18 years), family history of SA (yes; no), delay between the event and declaration to an adult, parental validation (yes; no). An ‘SA score’ of intensity was defined according to four SA variables considered to be ‘principal’ in the literature (nature, number, age of onset, relationship to the aggressor) with one point assigned for each item (penetration, repetition, age of onset ≤ 12 years, intrafamilial character). Once the score was measured for each patient, two sub-groups were formed: SA severity score 0 = 0, 1 or 2 points or 1 = 3 or 4 points.-Medical data: Number of suicide attempts, symptomatology, diagnosis, personal and family psychiatric history, overall level of functioning (measured by the Global Assessment of Functioning scale or GAF scale from the DSM-IV-TR).-Data related to care: Number and duration of hospitalizations.-Socio-judicial data: History of maltreatment (MT). A mistreatment score or ‘MT score’ was defined as the sum of other abuse reported by patients and/or observed by the team during hospitalization, with a point awarded for each abuse suffered (emotional abuse, physical abuse, neglect).

### 2.1. Statistical Analysis

Comparative analyses were performed using R.4.0.1 software. The association between the different variables was analyzed using statistical tests: Chi-2 test for categorical variables, Fisher’s exact test for dimensional variables, Pearson correlations. Then, we performed a multivariate linear regression including the variable to be explained ‘Time to disclosure’ and as explanatory variables ‘Sexual abuse score’ and ‘Maltreatment score’.

### 2.2. Ethics

All procedures were conducted in accordance with the fundamental ethical principles of the Declaration of Helsinki and in accordance with the ethical standards of the Hospital and University. The data processed were completely anonymous after extraction from medical records at discharge. The sample of hospitalised adolescents is a population admitted for free care at the public hospital, and therefore is not subject to any social discrimination. The psychiatric care was conducted as usual for all patients.

## 3. Results

### 3.1. Prevalence and Incidence

In the unit, which has an active line of approximately 120 patients per year, the prevalence of child SA calculated over the 4 years is 28.5% and the incidence is 21.1% (i.e., 21 new cases per year per 100 patients).

### 3.2. Description of the Study Population

#### 3.2.1. General Description

The study sample is composed of 100 patients, including 92 girls and 8 boys, aged 13 to 17 years old, 53% of whom had separated parents. A third (36%) of families were represented by doctoral level professions, 46% by intermediate professions, and 18% by temporary, seasonal, or per diem employment. This distribution is explained by the geographical sector corresponding to a more affluent population than the national average.

#### 3.2.2. Characteristics of the Sexual Abuse

Among the 100 files studied, the characteristics of the abuse are as follows:-Forty-nine percent of patients reported having suffered a single SA, 18% having suffered two sexual abuses, and 33% having suffered repeated sexual abuse (≥3).

Bearing in mind that each patient may be affected by several abuses,

-The age of SA reported is between 13 and 17 years old in 46% cases, 37% between 6 and 12 years old, and 17% between 0 and 6 years old. Fourteen patients revealed that they had been sexually abused in childhood and then again in adolescence.-One or more sexual touching were reported in the majority of cases (68%), 48% of cases mentioned one or more rapes, 31% reported having been victims of other violence (pornography, exhibition, threats). Only four patients were not affected by sexual touching or rape.-The vast majority of patients reported having been abused by a male individual (98 cases), it was an adult in 61 cases, an adolescent in 37 cases, and a child in 9 cases. A woman was the named aggressor in five situations.-The named aggressor was intrafamilial in 45% of cases and extrafamilial in 55% of cases. In cases of SA in childhood (≤12 years), the named aggressor was intrafamilial in 70% of cases.-In the vast majority of situations, the young person is the one who reveals the SA (88%). At the time of disclosure, the parents validate the statements in 63% of cases (53% for intrafamilial abuse and 72% for extrafamilial abuse). The time between the onset of SA and its disclosure varies between 1 day and 9 years, and extends on average over 3 years.-At the family level (second degree) a history of SA was found in 21% of cases.-Finally, the quantification of mistreatment (via the European Child Abuse and Neglect Minimum Data Set, 2015) showed that in the majority of cases (73%), one or more other mistreatments were reported: Emotional abuse (93%), physical abuse (40%), and neglect (93%).

#### 3.2.3. Clinical Characteristics

The average age at the time of hospitalization is 15.3 years. The average number of hospitalizations in psychiatry is three. The average cumulative duration of hospitalizations is 15 weeks (min = 0.5; max = 208), i.e., five times longer than the average patient over this period. The average GAF score is 32.2/100. Regarding personal history, the figures reported that 68% of the patients had already attempted suicide. Moreover, the vast majority of patients (67%) had previously mutilated themselves. Concerning family history, 65% of adolescents had one or both parents who had experienced psychological difficulties and 28% had a suicide family history.

#### 3.2.4. Symptoms and Diagnoses

The distribution of symptoms and diagnoses is depicted in Figure 1 and Figure 2. Patients most often present with several symptoms and several diagnoses could be attributed to each of them.

### 3.3. Characteristics of Sexual Abuse and Medical Severity

The correlations between characteristics of sexual abuse and medical severity as well as time to disclosure are reported in Table 1.

The medical severity of the patients was correlated with the age of the named author. The number of hospitalizations was correlated with the number of SA reported, as well as the intrafamilial and adult character of the named author. The number of suicide attempts was correlated with the number of SA reported. Finally, the time to disclosure was associated with the age of onset, the number of SA, the intrafamilial character, and parental validation.

### 3.4. Correlations between SA Score, MT Score, and Time to Disclosure

Multivariate analysis revealed that SA and MT scores were independently associated with time to SA disclosure. The time to disclosure was 2.13 times longer if the SA presented characteristics of particular gravity, and 2.42 times longer if there were additional mistreatments.

## 4. Discussion

### 4.1. General Features

Our study describes the extent and nature of SA phenomena reported in an adolescent population hospitalized in psychiatry, of well-off socio-economic level, and essentially female. Our study shows figures close to those described in the only study conducted on a comparable sample, in which 23.4% of 110 adolescents hospitalized reported sexual abuse [10]. However, in our study, the average age of care is around 15 years old, and this young age added to the time required for disclosure may be in favor of higher figures found in the samples or estimates for adults. For example, Zanarini et al. [23] described that 61.5% of patients with borderline personality disorders reported having been sexually abused, and 35% of subjects consecutively admitted to a general hospital after having made a suicide attempt reported a severe SA [24].

The low number of young men in this sample could be linked to the ratio of hospitalized young people (one boy/two girls) added to a male declaration lower than the female declaration in the general population [2,3]. The authors of these studies generally assume a bias related to additional taboos, leading to under-reporting in men compared to women. Population-based clinical measurement studies show that SA is also very frequent in men, and significantly higher than in the general population [25]. Therefore, of 125 consecutive male patients at an adult psychiatric outpatient clinic, 48% reported histories of sexual and/or physical abuse [25]. In our study, the ratio of males to females on the ward suggests that young adolescent males underreport SA even more than their adult counterparts. In addition, it is possible that the symptoms generated by SA are in favor of more externalizing disorders, leading them to juvenile justice and substance abuse program, rather than psychiatric units. In this sense, Romano et al. [26] showed that men waited an average of 15.4 years before sharing their experience, and that greater delay in disclosure predicted greater externalizing behaviors. These results suggest that efforts need to be undertaken to address the barriers that hinder men from disclosing their sexual abuse and to ensure that men are supported once they disclose.

Since CSA is often part of a pattern of polyvictimization it is difficult to isolate the effects of one type of trauma or abuse from the other. Very few studies have included more than one or two of these variables and have statistically controlled for their effects. Our results revealed an additional effect of other forms of maltreatment on the time to disclosure of SA (Table 2) [27,28]. This suggests that measuring the impact of all forms of victimization alongside CSA is warranted in order to fully capture the influence of violence and abuse, not only on the development of mental health outcomes, but also on disclosure. In this line, other studies observed specific but cumulative effects of maltreatment subtypes on psychopathology, for example, borderline one [23,29].

### 4.2. SA Characteristics and Clinical Severity

Several results of our study point toward SA affecting the clinical severity and the amount of care needed: The repetition of SA is correlated with a greater number of suicide attempts and hospitalizations in adolescence, whereas intrafamilial SA is related to the number of hospitalizations in adolescence. These results echo several studies that underline the impact of sexual violence on the medical severity of the victims, associating in them earlier onset of disorders, of poorer prognosis, responding less well to treatment and associating a high risk of suicidal repetition [15,30,31,32,33,34]. Victims of SA are more prone to abusing substances, to engaging in self-harm behaviors, and to attempting or committing suicide. Moreover, the mental health outcomes of CSA victims are likely to continue into adulthood as the link of CSA to lifetime psychopathology has been demonstrated [20]. A 23-year longitudinal study of the impact of intrafamilial sexual abuse on female development confirmed the deleterious impact of CSA across stages of life, including all of the mental health issues, but also hypothalamic–pituitary–adrenal attenuation in victims, as well as asymmetrical stress responses, high rates of obesity, and healthcare utilization [35]. In all cases, early assessment and intervention to offset the exacerbation and continuation of negative outcomes is highlighted, as symptoms can develop at a later age or may not be apparent at first. Our results complement previous studies by directly observing these outcomes in adolescence.

However, the nature of SA did not reveal any correlation with medical variables. It is possible that the overall intensity of SA reported in this study (at least touching or rape in 95% of cases) and the intensity of the mental disorders have reduced the possibility of revealing these correlations, which are nevertheless described in the literature [22,36]. Instead, the age of the attacker seems to play an important role both on the clinical severity of the patients and the number of their hospitalizations (Table 1). Indeed, SA perpetrated by an adult, in addition to representing relational violence, has a major impact on the representations related to the transgression of laws between generations, which thus questions the protective role of the adult, and also the reliability of the attachment, if the aggressor is a close person [37]. Our results are consistent with the work of Steine [38] who emphasized the importance of cumulative childhood sexual abuse as a predictor of suicidal ideation.

### 4.3. Clinical Polymorphism

The descriptive clinical data report a symptomatic polymorphism (Figure 1 and Figure 2), mixing mood disorders, behavioral disorders (including eating disorders), dissociative symptoms, symptoms of post-traumatic stress disorder, and somatizations. Literature data have reported that the consequences associated with SA are very diverse and do not allow for the establishment of a clinical profile [39]. There is evidence that CSA is now considered as a general, nonspecific risk factor for psychopathology (including psychologically, behaviorally, and sexually related problems and later revictimization as well as somatizations) [11]. However, a link is particularly well established between a history of SA and the occurrence of self-harm, NSSI, and suicidal behaviors [29,34,40], which could support the very high proportion of these behaviors in our cohort (two-third of patients). It is now considered that CSA makes both direct and indirect contributions to suicidal behavior. Psychopathologies, such as post-traumatic stress disorders, personality disorders (including impulsivity), and mood disorders may mediate the relationship between CSA and suicidal behavior. It is a complex process involving psychopathology, maladaptive personality features, and the direct contribution of CSA itself [41].

The high prevalence of sadness and anxiety symptoms in our population echoes the high levels of mood disorders, such as major depressive episodes, which are described in cohorts of children and adolescents who have been sexually abused [22]. The prominence of these symptoms, in front of those directly resulting from a post-traumatic state (reexperiencing, hypervigilance, derealization, etc.) interrogates the psychopathology of trauma. Links between traumatic psychopathology and thymic disorders (not only anxiety) have been described and have led to discussion of common psychopathological pathways underpinning both of these clinical processes [42,43]. The despair linked to the powerlessness of the victim of SA as well as the profound alteration of the self-esteem probably contribute to this sadness in the foreground. Moreover, in children and adolescents, trauma shows particularities linked to the precocity of its occurrence during development, thus leading to the development of symptoms that are broader than the classical post-traumatic register: The complex trauma describes a clinical picture that associates dysregulation of affects, profound alteration of self-esteem, and insecurity of attachment that is often marked [44]. The diversity of symptoms observed in this sample with a high level of trauma suggests a significant preeminence of diagnostic forms that would be closer to complex trauma.

### 4.4. Disclosure, Parental Support

The correlation observed in our study between the age of onset of SA and the time of disclosure (Table 1) suggests that children wait until adolescence to reveal SA, whereas an onset in adolescence is revealed faster. These results echo recent observations of patients’ accounts who finally declare to have been aware of their abuse since childhood but have waited to be more able to protect themselves, to protect their siblings or to feel capable of being believed [45]. This time to disclosure is significantly longer if the aggression is intrafamilial (Table 1), probably linked to various factors, foremost among which is loyalty to family cohesion [46,47]. Intrafamilial aggression is also associated with more frequent hospitalizations (suggesting an effect of escape from familial danger through hospitalization) and questions the existence of maintaining the link with the aggressor as a factor of recurrence of hospitalizations. Moreover, it questions the place of the medical and child protection response in these situations. Indeed, these results, as well as clinical practice, suggest a particular severity of these situations in terms of psychological disorders, but also a difficulty in organizing a return to the family home in this context, which could explain re-hospitalizations if the child protection services do not take over with a placement of the adolescent. Therefore, it is crucial to be able to articulate the responses of these two professional fields in order to avoid a hospital response that is made by default of another response.

Parental validation is a characteristic very often mentioned in the literature as a factor favorably influencing the state of health of young people with SA as well as their interpersonal functioning [21,48,49,50]. A systematic review highlighted that the child’s decision to disclose an experience of SA was affected by six factors: Fear of what will happen, other’s reaction, fear of disbelief, emotion and impact of the abuse, an opportunity to tell, concern for self and others, and feelings toward the abuser [51]. These factors show the extent to which the child anticipates the possibility of validation by those around him of his revelation. According to our results of the regression analysis, the more extreme the sexual violence (high SA score), the more the victim delays disclosure, and this is independent of other mistreatments (which also seem to play a cumulative role). This implies that the younger the child is and the more the abuse is repeated by someone close to him, the more difficult it will be for him to reveal the abuse. This seems logical since the relationship of dependence to this person is major in these situations. The time required for disclosure appears to be proportional to the quality of the response to disclosure estimated by the child or adolescent, i.e., the quality of the relationship of trust. In this sense, research results also show that a secure attachment relationship significantly predicts the consequences of SA in children victims following disclosure [52]. Therefore, the relationship of the child or adolescent to his or her caregiver(s) is essential, both before and after the disclosure. Moreover, the observation in our study of less parental validation in situations of intrafamilial sexual abuse suggests the specificity of the denial mechanisms in place in these situations, beyond the question of relational trust [53].

Finally, the difference between our observation of a history of SA in relatives (25%) and the literature which reports an average prevalence of 50% of SA in the childhood of mothers of sexually abused children [54] can be explained by the reluctance to bring up family history in the context of child care, despite exploration by clinicians.

### 4.5. Limitations

Among the limitations of this study, we first note a traceability bias in the medical records that is the source of some missing data. Second, the collection of diagnoses was not standardized, but was carried out according to the DSM five criteria. Moreover, we lacked data concerning the legal consequences and their impact on symptoms evolution. It is reported in the literature that children involved in long, unresolved court proceedings involving numerous testimonies showed slower recovery than those who did not have a court case or those for whom the case ended faster through a conviction or plea bargain. It would be useful to be able to describe the follow-up to legal procedures in these complex clinical situations in future studies.

## 5. Conclusions

This research underlines the major place of SA in adolescent psychiatry. It echoes and pursues some data described in adults; and it constitutes a basis of comparison for future studies. Moreover, it suggests that the prevention of SA occurrence and its repetition as well as its consequences (particularly linked to its late disclosure or the lack of support of the young person) represents powerful arguments for the prevention of mental disorders in adolescence. Indeed, the duration of hospitalizations, their repetition, as well as the repetition of suicide attempts, are statistically in favor of a history of SA. Therefore, SA history should be more actively searched for in the adolescent daily practice, especially when faced with suicide attempts, and regardless of the adolescent’s clinical diagnosis (given the diversity of symptoms observed). As a result, increasing the detection rate of CSA within the context of child and adolescent psychiatry represents a significant secondary prevention strategy. Moreover, these results provide arguments for improving the education of the general population, medical community, mental health providers, and other frontline staff, in order to assist in mobilizing earlier interventions that, in turn, can address these risk factors resulting in decreased SA and suicidal behavior. Increased access to care, including mental health services before emergency and hospitalization, may improve outcomes as well. Furthermore, the results suggest that the development of specialized emergency services for consequential needs related to sexual abuse, would make it possible to prevent suicidal behaviors and mental disorders in adolescence.

## Figures and Tables

**Figure 1 ijerph-20-01071-f001:**
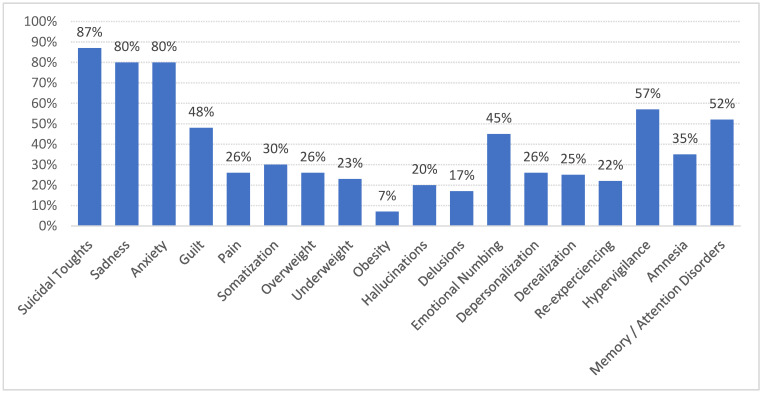
Distribution of symptoms.

**Figure 2 ijerph-20-01071-f002:**
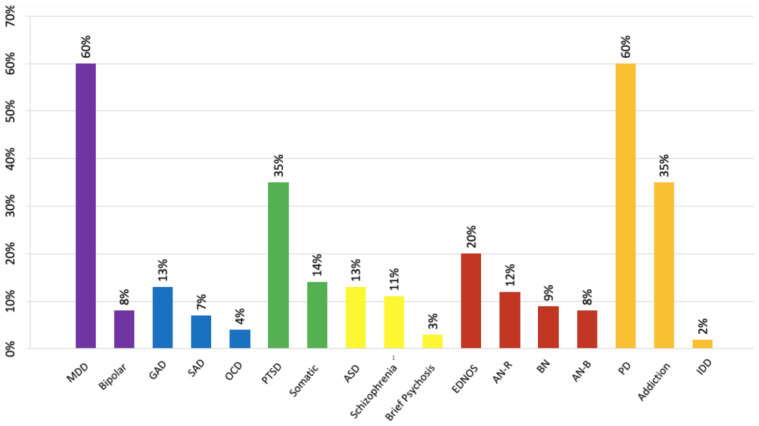
Breakdown of diagnoses. MDD: Major Depressive Disorder; Bipolar: Bipolar Disorder; GAD: General Anxiety Disorder; SAD: Social Anxiety Disorder; OCD: Obsessive-Compulsive Disorder; PTSD: Post-Traumatic Stress Disorder; Somatic: Somatic Symptom and Related Disorders; ASD: Autism Spectrum Disorder; Brief Psychosis: Brief Psychotic Disorder; EDNOS: Eating Disorder Not Other Specified; AN-R: Anorexia Nervosa Restrictive Type; BN: Bulimia Nervosa; AN-B: Anorexia Nervosa-Binge-Eating/purging Type; PD: Personality Disorders; Addiction: Addictive Disorders; IDD: Intellectual Developmental Disorder.

**Table 1 ijerph-20-01071-t001:** Correlations between characteristics of sexual abuse and medical severity as well as time to disclosure (GAF: Global Assessment of Functioning scale; SA: Sexual abuse).

	GAF Score	Number of Hospitalizations	Number of Suicide Attempts	Time to Disclosure
**Age of the victim**				
≥13 years (*n* = 46%)	34.2 (±7.43)	2.88 (±1.86)	1.79 (±2.20)	1.08 (±1.64)
<13 years (*n* = 54%)	31.6 (±8.61)	3.18 (±2.02)	1.58 (±1.47)	4.45 (±3.36)
	*p* = 0.11	*p* = 0.46	*p* = 0.61	*p* < 0.001
**Number of SA**				
Single (*n* = 49%)	34.1 (±7.4)	2.70 (±1.83)	1.35 (±1.75)	2.16 (±2.89)
Recurrent (*n* = 51%)	31.9 (±8.48)	3.38 (±1.90)	2.28 (±2.0)	4.02 (±3.16)
	*p* = 0.2	*p* = 0.05	*p* < 0.01	*p* < 0.01
**Nature of SA**				
Without penetration (*n* = 52%)	32.8 (±8.32)	3.12 (±1.98)	1.83 (±2.30)	2.96 (±3.17)
With penetration (*n* = 48%)	32.2 (±8.21)	3.04 (±2.0)	1.70 (±1.82)	3.22 (±3.29)
	*p* = 0.71	*p* = 0.86	*p* = 0.77	*p* = 0.73
**Relationship to the perpetrator**				
Extrafamilial (*n* = 55%)	33.6 (±7.99)	2.6 (±1.6)	1.56 (±2.10)	1.70 (±2.46)
Intrafamilial (*n* = 45%)	31.3 (±8.43)	3.53 (±2.26)	1.95 (±1.91)	4.24 (±3.40)
	*p* = 0.18	*p* = 0.025	*p* = 0.34	*p* < 0.001
**Age of the named perpetrator**				
<18 years (*n* = 43%)	36.1 (±7.14)	2.42 (±1.3)	1.36 (±1.34)	2.95 (±3.17)
≥18 years (*n* = 57%)	30.6 (±8.23)	3.31 (±2.2)	1.97 (±2.29)	3.17 (±3.47)
	*p* < 0.01	*p* = 0.016	*p* = 0.11	*p* = 0.7
**Parental validation**				
Yes (*n* = 63%)	33.2 (±7.85)	2.92 (±2.07)	1.90 (±2.31)	2.37 (±2.81)
No (*n* = 37%)	32.3 (±8.96)	3.37 (±1.8)	1.59 (±1.37)	4.22 (±3.52)
	*p* = 0.74	*p* = 0.11	*p* = 1	*p* = 0.037
**Family history of SA**				
No (*n* = 79%)	33.3 (±8.0)	2.83 (±1.91)	1.48 (±1.71)	2.96 (±3.18)
Yes (*n* = 21%)	31.8 (±5.11)	2.5 (±1.2)	1.89 (±2.40)	2.01 (±2.25)
	*p* = 0.3	*p* = 0.86	*p* = 0.6	*p* = 0.37

**Table 2 ijerph-20-01071-t002:** Multivariate analysis of time to disclosure as a function of SA and MT scores (SA: Sexual abuse; MT: Maltreatment).

		Coefficients	*p*	Global *p*
SA score	1 vs. 0	2.13 [0.760; 3.34]	<0.001	<0.001
MT score	1 vs. 0	0.285 [−1.09; 1.74]	0.78	<0.01
	2 vs. 0	0.776 [−0.543; 2.10]	0.28	-
	3 vs. 0	2.42 [0.944; 3.90]	<0.01	-

## Data Availability

Available online: https://doi.org/10.6084/m9.figshare.21261348.v1 (accessed on 1 December 2022).

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
