# Peer review of "Clinical Stakes of Sexual Abuse in Adolescent Psychiatry"

_ijerph, 2023, doi:10.3390/ijerph20021071_

Round 1

Reviewer 1 Report

A worthy and important topic for Child Psychiatry, perhaps especially in inpatient settings, but one would hope that better screening and detection at the outpatient (or even school counseling) level could help to mitigate against these very high rates.  My concerns are relatively minor, and I hope that the authors will not have too much difficulty implementing them:

1. First, a few English language edits:

- p. 2, ln. 48-49 ("a central element of his future") -- A bit poetic for a scientific paper.  Perhaps "predict better [longitudinal] outcomes" instead?

- p. 3, Sec. 3.2.1 - By "intellectual professions" do you mean "doctoral level professions"?  By "precarious jobs", do you mean "temporary, seasonal, or per diem employment"?  By "well-off", do you mean "affluent"?  Also, without some context, I was curious about the grouping of "manual workers" with seasonal workers unless these too were also temporary employees or subcontractors?  (In the US, manual labor is often unionized and paid quite well).  

- Figure 2: "Underweight" would be preferable to "Thinness" in the X-axis (and would better group with the other clinical weight categories included in the graph).  Also, since you used DSM-5 criteria to create the symptomatic categories, "Emotional Numbing" might be preferable to "Emotional Detachment" so as not to confuse with dissociation.

- Page 7, Section 4.5: The title "Limitations" is sufficient and more appropriate and accessible for English-language journal readers

2. Page 6, ln 180-185: Curious about whether you've looked at the literature on delinquency and acting out among male survivors of SA, and might that in part help to explain the disproportionately high percentage of females in your sample.  (In other words, are males coming to the attention of juvenile justice and substance abuse programs, whereas females are coming to psychiatric units)?

3. Page 2, Section 4.3 (Clinical Polymorphism): I would direct the authors to research on the structure of psychopathology in PTSD for their interest and further exploration/explanation of their very interesting findings
(e.g., Cox, Clara & Enns, 2002: https://onlinelibrary.wiley.com/doi/abs/10.1002/da.10052; Miller et al., 2008: https://www.ncbi.nlm.nih.gov/pmc/articles/PMC3561690/; Kira et al., 2017: https://www.scirp.org/html/8-6902310_81114.htm?pagespeed=noscript)

4. p. 7, ln 221: "...deferred action in mental dynamic during puberty" - Are you referring to the latency phase of Freud's psychodynamic model (or perhaps Blos's extension of Freudian theory into adolescence)?  While interesting, I'm not sure it is applicable here.  Lines 222-224 are better grounded in the literature and make your case just as well.  

Reviewer 2 Report

This is an excellent addition to the literature. This study examines the psychiatric correlates of childhood sexual abuse in an adolescent population. In contrast, most prior studies examined correlates of sexual abuse history in adult populations. It would be of considerable value to conduct a replication study in a different population. The population in this study is more affluent than the general population. It is possible, theoretically, that this might be significant. Additionally, I concur with the authors that it would be useful to describe the follow-up to legal procedures.

Some minor requests for clarification follow.

Page 1. “Child Sexual Abuse (CSA) is thought to be associated with 47% of psychiatric disorders appearing in childhood and about 30% of disorders appearing in adulthood…”

Page 5. The comparison of our data with those of the literature shows relatively comparable rates of sexual abuse in the psychiatric population (around 25%)…”

I am trying to understand the discordance between 47% and 25%. Is a distinction being made here between childhood sexual abuse and sexual abuse throughout the life span?

A clarification concerning the Global Assessment of Functioning might be helpful. Was the GAF as given in the DSM-IV-TR used? Alternatively, there is Hall, R.C. (1995) Global assessment of functioning: a modified scale. Psychosomatics 36(3), 267-275,  which gives a more structure scoring system, or Shaffer, et al. (1883). A Children’s Global Assessment Scale. Archives of General Psychiatry. 40, 1228-1231.

Reviewer 3 Report

Dear Authors,

This issue is crucial to invest age. I appreciate your work. Since this issue is sensitive, and patients under 18 and have mental disorders are vulnerable, accessing patients' data, you should add an ethical consideration section and explain where and when you have obtained ethical approval and written consent both adolescents and their legal guardian.

There are some points that I would like to highlight;

- There are different terms such as psychologic and psychiatric disorders. That could be converted to mental disorders which is a MeSH term. Also, you could add the keywords mental disorders.

- 100 adolescents participated in this study. Is there any sample calculation? 100 is the very general sample size. Please add the details.

- All the references are older than 10 or 20 years. I recommend using the current studies in this manuscript. 

The introduction, method, and discussion part were well written. The results and graphs are clear. I appreciate your effort.

I wish you success in your work. 

Reviewer 4 Report

1.          In the section of introduction, I suggest the author should provide more literature review about the meaning, development, or practices about the clinical stakes, sexual abuse, adolescent psychiatry, and such relationships.

2.          In the section of materials and methods, I suggest the author should provide more details about the methodology and its practical considerations or strategies.

3.          In table 1, the author provided the correlations between characteristics of sexual abuse and medical severity as well as time to disclosure. But I did not find some theoretical discussions about this issue. I suggest the author should provide more literature review or hypothesized consideration about this concern.

4.          In the section of conclusion, I suggest the author should provide more information about the theoretical reflections, practical suggestions, and future suggestions based on the statistical results.

Round 2

Reviewer 4 Report

The author mostly incorporated my suggestions and revised the manuscript. The manuscript could be published in this form for the journal.